# Micro-Macro Coupled Koopman Modeling on Graph for Traffic Flow Prediction

**Bairan Xiang, Chenguang Zhao & Huan Yu**
Thrust of Intelligent Transportation
The Hong Kong University of Science and Technology (Guangzhou)
`{bxiang107,czhao704}@connect.hkust-gz.edu.cn`
`huanyu@hkust-gz.edu.cn`

## Abstract

Traffic systems are inherently multi-scale: microscopic vehicle interactions and macroscopic flow co-evolve nonlinearly. Microscopic models capture local interactions but miss flow evolution; macroscopic models enforce aggregated consistency yet overlook stochastic vehicle-level dynamics. We propose Micro–Macro Coupled Koopman Modeling (MMCKM), which lifts the coupled dynamics to a high-dimensional linear observation space for a unified linear-operator representation. Unlike grid-based discretizations, MMCKM adopts a vehicle-centric dynamic graph that preserves microscopic perturbations while respecting macroscopic conservation laws by discretizing PDEs onto this graph. At the micro scale, scenario-adaptive Koopman evolvers selected by an Intent Discriminator are designed to model vehicle dynamics. A Koopman control module explicitly formulate how flow state influences individual vehicles, yielding bidirectional couplings. To our knowledge, this is the first work to jointly model vehicle trajectories and traffic flow density using a unified Koopman framework without requiring historical trajectories. The proposed MMCKM is validated for trajectory prediction on NGSIM and HighD. While MMCKM uses only real-time measurement, it achieves comparable or even higher accuracy than history-dependent baselines. We further analyze the effect of the operator interval and provide ablations to show the improvement by intent inference, macro-to-micro control, and diffusion. Code and implementation details are included to facilitate reproducibility.

## 1 Introduction

Traffic flow modeling is a fundamental challenge in intelligent transportation systems, since it requires simultaneously understanding of intrinsically coupled microscopic vehicle behaviors and macroscopic flow dynamics. Individual vehicle maneuvers aggregate to form traffic patterns, while macroscopic states constrain and influence microscopic driving decisions. The bidirectional coupling creates a complex nonlinear system (Wang et al., 2024). Existing methods typically adopt either a microscopic or macroscopic perspective, failing to capture the critical cross-scale interactions. Microscopic-based approaches model individual vehicle dynamics through causal temporal inference (Mukherjee et al., 2020; Messaoud et al., 2020) or multi-agent interactions on spatial domain (Rahmani et al., 2023; Shi et al., 2024; Gao et al., 2025). While these methods capture local behaviors and stochastic events, they struggle to maintain global flow consistency and scale poorly with vehicle numbers. On the other side, macroscopic models use partial differential equations (PDEs) such as the Lighthill-Whitham-Richards (LWR) model to ensure conservation laws and flow continuity (Hu et al., 2022; Mahjourian et al., 2022), but they are deficient at responding to individual vehicle events that critically influence traffic evolution (Haghighi & El Amine Hamri, 2024; Rowan et al., 2025).

Despite recent advances, a fundamental gap remains: few existing methods simultaneously bridge microscopic and macroscopic traffic while maintaining computational tractability and physical interpretability. Recent efforts to bridge micro-macro scales through game-theoretic frameworks(Huang et al., 2020) or kinematic limits(Cristiani & Sahu, 2016) offer theoretical insights, but they either assume homogeneous drivers or asymptotic regimes, limiting their real-world applicability. (Lat-

Figure 1: We propose a Micro-Macro Coupled Koopman Modeling (MMCKM) framework that lifts microscopic vehicle trajectories and macroscopic flow evolution into linear observation spaces, enabling unified prediction. The traffic environment is formulated as a directed graph. At the macroscopic level, vehicle-centric graph discretization captures how microscopic perturbations affect wave propagation. At the microscopic level, scenario-adaptive Koopman operators and Koopman control incorporate macroscopic flow conditions into vehicle dynamics.

tanzio & Piccoli, 2010; Bellomo et al., 2014; Wang et al., 2024; Fan et al., 2025). The fundamental challenge remains how to model the nonlinear, bidirectional coupling in a unified, computationally tractable framework.

We propose a Micro-Macro Coupled Koopman Modeling (MMCKM) framework that leverages Koopman operator theory to transform nonlinear multi-scale dynamics into high-dimensional linear observation spaces, enabling unified prediction through linear operators as in Fig. 1. Our key insight is that both microscopic vehicle trajectories and macroscopic flow evolution can be lifted into respective observation spaces where their dynamics become approximately linear. Besides, the Koopman operator exhibits the Markovian property when observation functions are time-invariant (Wu & Noé, 2020; Kostic et al., 2022; Tafazzol et al., 2024). Therefore, our framework relies solely on current state information without requiring historical tracking or continuous object detection. The bidirectional coupling is achieved through the following mechanisms: microscopic vehicle events influence macroscopic flow through a diffusion term added to the LWR model, capturing how individual vehicle perturbations propagate through traffic flow. Unlike existing approaches that discretize traffic flow on fixed spatial grids of the Euclidean coordinate, we propose a vehicle-centric graph discretization on the Lagrangian coordinate that preserves microscopic perturbations while maintaining macroscopic conservation laws. For the microscopic dynamics, we design a scenario-adaptive Koopman operator selection mechanism to capture vehicle dynamics in different driving scenarios. And macroscopic flow states affect microscopic vehicle dynamics through Koopman control, where flow conditions serve as external inputs to affect individual vehicle dynamics.

Our framework makes three major contributions:

1. **Vehicle centric PDE on graphs**: We derive an advection–diffusion evolution on a dynamic vehicle graph with skew-symmetric advection and positive semi-definite diffusion, ensuring energy-preserving advection and nonnegative diffusion process, with a constructive parameterization for antisymmetry.

2. **Unified history-free Koopman modeling**: We show how macro/micro observables can be evolved by time-invariant Koopman operators, enabling accurate trajectory prediction from a single snapshot; we align Koopman spectra with graph-PDE spectra to improve stability and interpretability.

3. **Physics-guided multi-regime micro dynamics**: A lightweight intent discriminator selects among parameter-bounded Koopman evolvers; a Koopman-control path injects macro flow with ISS-style bounds on control/output, reducing long-horizon drift.

Beyond predictive performance, our framework provides explicit insights through learned edge weights that quantify interaction intensities between vehicles. These weights evolve dynamically to reflect changing driving conditions, offering interpretable measures for downstream decision-making modules in autonomous driving systems, a capability unique to our vehicle-centric graph formulation.

## 2 BACKGROUND KNOWLEDGE

### 2.1 KOOPMAN OPERATOR THEORY & KOOPMAN CONTROL THEORY

The Koopman operator provides a powerful framework for analyzing nonlinear dynamical systems through linearization in an observation space. Considering a discrete-time nonlinear dynamical system:

$$x_{t+1} = f(x_t), \quad x_t \in \mathbb{R}^n \tag{1}$$

By Koopman operator theory, there exists an infinite-dimensional Hilbert space of observable functions:

$$g := \{\phi_i\}_{i=1}^\infty, \quad \mathbb{R}^n \to \mathbb{R} \tag{2}$$

where system dynamics become linear. The Koopman operator $\mathcal{K}$ acts on observables as:

$$\mathcal{K}g = g \circ f \tag{3}$$

where $\mathcal{K}$ is a linear (though infinite-dimensional) operator. In the lifted observation $z_t = g(x_t) = [\phi_1(x_t), \phi_1(x_t), ...]^\top$, the evolution becomes:

$$z_{t+1} = \mathcal{K}z_t \tag{4}$$

The original state can be recovered through reconstruction functions $\psi : z \mapsto x$.

**Finite-dimensional approximation via DMD**: While Koopman operator requires infinite dimensions, Dynamical Mode Decomposition (DMD) enables finite-dimensional approximation suitable for practical applications(Brunton et al., 2016). DMD identifies a finite set of modes that capture the dominant dynamics, with each mode characterized by an eigenvalue $\lambda_j$ whose real part determines growth/decay rates and imaginary part captures oscillation (Avila & Mezić, 2020).

**Neural network parameterization**: Recent advances leverage neural network's universal approximation capabilities to learn both observable functions $\phi : \mathbb{R}^n \mapsto \mathbb{R}^d$ and reconstruction functions $\psi : \mathbb{R}^d \mapsto \mathbb{R}^n$(Lusch et al., 2018). The key insight is to learn a representation where the Koopman operator becomes approximately diagonal:

$$x_{t+1} = \psi(K\phi(x_t)) \tag{5}$$

where $K \approx \text{diag}(\lambda_1, ...\lambda_d)$. The diagonalization enables efficient evolution:

$$z_{t+1}^j = e^{\lambda_j} z_t^j \tag{6}$$

providing both computational efficiency and interpretability through modal decomposition.

**Extension to controlled system**: For systems with external inputs:

$$x_{t+1} = f(x_t, u_t) \tag{7}$$

Koopman control theory introduces an actuation operator $\mathcal{B}$ that lifts control inputs into the observation space(Proctor et al., 2018; Strässer et al., 2023):

$$z_{t+1} = \mathcal{K}z_t + \mathcal{B}u_t \tag{8}$$

This formulation maintains linearity in the observation space while accommodating external influences. Crucially, the framework imposes minimal constraints on the control structure, allowing us to model abstract influences, such as macroscopic traffic flow effects on individual vehicles, as control inputs. This flexibility is particularly valuable for multi-scale systems where cross-scale interactions lack explicit mathematical forms but significantly impact dynamics.

The Markovian property of the Koopman evolution, when observation functions are time-invariant, enables prediction using only current state information—eliminating the need for historical tracking that burdens conventional sequence-based methods.

### 2.2 MACROSCOPIC TRAFFIC FLOW MODELS

Macroscopic models treat traffic as a continuum fluid, with the LWR model serving as the fundamental first-order theory. The traditional LWR model describes traffic density evolution through a conservation equation describing the fluid propagation dynamics:

$$\frac{\partial \rho}{\partial t} + \nabla \cdot \mathbf{Q}(\rho) = 0 \tag{9}$$

where $\rho(x,t)$ represents traffic density and $\mathbf{Q}(\rho) = \rho\mathbf{v}(\rho)$ is the flux function with $\mathbf{v}(\rho)$ denoting the density-dependent velocity and wave propagation speed field direction which can be viewed as an advection process.

Traditional PDE discretization methods for traffic flow work on the Euclidean coordinate and divide roads into fixed spatial cells. This approach fundamentally limits their ability to capture vehicle-level perturbations. Stochastic behaviors are averaged within each cell, which eliminates the high-frequency dynamics that is crucial for understanding traffic flow. Our work addresses this limitation by introducing a Lagrangian discretization on vehicle-centric graphs.

## 3   PROBLEM FORMULATION

We address the multi-scale traffic flow modeling problem where both individual vehicle dynamics and macroscopic traffic density evolution are jointly considered. Unlike traditional models that focus solely on vehicle behaviors or flow modeling, our framework captures the bidirectional coupling between microscopic vehicle dynamics and macroscopic flow patterns.

We assume that the ego vehicle can only get measurements of surrounding vehicles within a detection range $r_{\max}$. At each time step $t$, we represent the traffic system as a dynamical weighted directed graph $\mathcal{G}_t = (\mathcal{V}_t, \mathcal{E}_t, \mathcal{W}_t)$, where:

- Nodes $\mathcal{V}_t$ represent vehicles. For a vehicle $i$, its state $x_t^i$ includes position $p_t^i \in \mathbb{R}^2$, velocity $v_t^i \in \mathbb{R}^2$, lane ID $l_t^i$ and vehicle size $s^i \in \{0,1\}$ with $s^i = 0$ being large vehicles whose length is longer than 6.5 m and $s^i = 1$ being small vehicles. For the ego vehicle, we use the notation as $x_t^e = [p_t^e, v_t^e, l_t^e, s_t^e]$.

- Edges $\mathcal{E}_t$ describe spatial interaction between vehicles. We construct using k-nearest neighbors (k-NN) based on Euclidean distance to balance computational efficiency with interaction coverage. We use incidence matrix $B_t$ to represent the interaction between them. For two vehicles $i$ and $j$, we set $B_{t,ij} = 1$ if there is an edge from $v_j$ to $v_i$ is interaction between them and $B_{ij} = 0$ otherwise.

- Edge weights $\mathcal{W}_t$ quantifies vehicle interaction intensity and traffic flow propagation efficiency. To reflect the propagation of traffic flow and the effect of vehicle interaction on traffic flow, we use two edge weights and learn them from node features.

Usually, the graph topology is described by graph Laplacian operator $L = BWB^\top = D - A$, where $D$ is the degree matrix and $A$ is the adjacency matrix. The graph Laplacian $L$ has a conjugate edge Laplacian $L_e = B^\top WB$ which has the same non-zero eigenvalues $L$. This formulation enables us to express traffic dynamics through graph operators that preserve physical laws.

We adopt Koopman-based approach to lift the nonlinear coupling micro-macro system to a higher-dimensional linear observation space. A critical advantage of Koopman-based approach is its Markovian property, which enables us to predict using only current state information $\mathcal{G}_t$ without requiring historical trajectories. The model predicts the ego vehicle future trajectory position and traffic density evolution over $T_f$ prediction.

## 4   METHODOLOGY

In this paper, we model the cross-scale traffic dynamics. The framework is illustrated in Fig.1.

### 4.1   TRAFFIC FLOW EVOLUTION ON GRAPH

Traditional spatial discretization fundamentally cannot capture how individual vehicle perturbations propagate through traffic. We pioneer a Lagrangian approach that discretizes PDEs directly onto vehicles as graph nodes. This is not merely a change of coordinates—it fundamentally preserves information that spatial methods inherently lose. We give the detailed proof in Appendix.A.1 which is one of the core innovation in this paper. The evolution on graph is:

$$\dot{\boldsymbol{\rho}} = -C^{\mathrm{adv}}\boldsymbol{\rho} + L^{\mathrm{diff}}\boldsymbol{\rho}, \quad C^{\mathrm{adv}} = B^\top W^{\mathrm{adv}}B, \quad L^{\mathrm{diff}} = B^\top W^{\mathrm{diff}}B \tag{10}$$

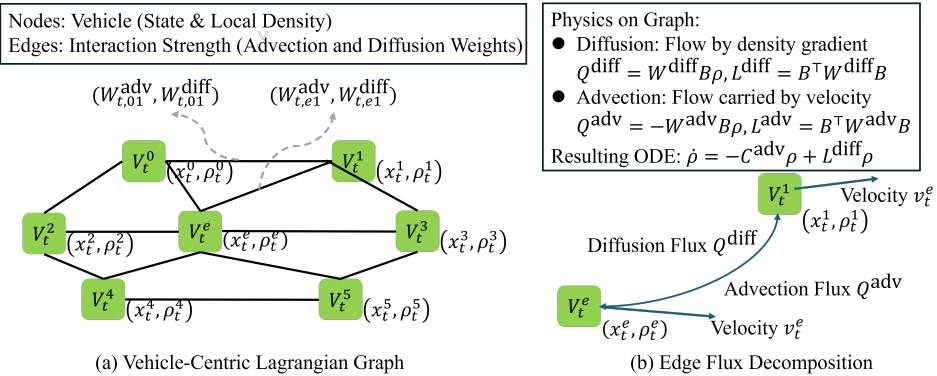

(a) Vehicle-Centric Lagrangian Graph      (b) Edge Flux Decomposition

Figure 2: Illustration of traffic flow evolution on vehicle-centric Lagrangian Graph. (a) Lagrangian discretization: Unlike traditional Eulerian methods that discretize space into fixed grids, we treat vehicles as dynamic nodes $\mathcal{V}_t$ in the graph $\mathcal{G}_t$. The mesh moves with the traffic flow, allowing the model to preserve high-frequency microscopic perturbations. (b) Edge flux decomposition: The advection flux captures velocity-induced transport via a skew-symmetric $C^{\text{adv}}$, while the diffusion flux captures density-gradient interactions via a PSD operator $L^{\text{diff}}$.

where $\boldsymbol{\rho} = [\rho_1, ..., \rho_N]$ is a vector with $\rho_i$ being the density at the position of vehicle $i$, $N$ is the number of vehicle, $C^{\text{adv}}$ is the advection operator which is an antisymmetric matrix describing the traffic flow, and $L^{\text{diff}}$ is the diffusion operator which is a positive semi-definite (PSD) matrix representing the perturbation from surrounding vehicles. We use two edge weight matrices $W^{\text{adv}}$ and $W^{\text{diff}}$ to denote the advection and diffusion coefficients respectively. This formulation is fundamentally different from previous work that discretize space on the Euclidean coordinate and calculate density as average traffic state within cells. Instead, we discretize on the Lagrangian coordinate and get density values around each vehicles. Therefore, how each vehicle affects flow propagation is explicitly formulated. Specifically, we use edge weights $W^{\text{adv}}$ and $W^{\text{diff}}$ to explicitly encode vehicle-to-vehicle interaction strengths, providing interpretable measures that are unavailable in grid-based methods.

According to graph spectrum thoery, if $L^{\text{diff}}$ and $C^{\text{adv}}$ are commute, i.e., $L^{\text{diff}} C^{\text{adv}} = C^{\text{adv}} L^{\text{diff}}$, they can be diagonalized by the same eigenvectors $U$:

$$\begin{aligned}
\tilde{L}^{\text{diff}} &:= U^* L^{\text{diff}} U = \text{Diag}(\eta_1, \cdots, \eta_N), \\
\tilde{C}^{\text{adv}} &:= U^* C^{\text{adv}} U = \text{Diag}(\text{j}\xi_1, \cdots, \text{j}\xi_N),
\end{aligned} \tag{11}$$

where $\text{j}^2 = -1$ is the imaginary unit, $\eta$ are the eigenvalues of $L^{\text{diff}}$ and $\text{j}\xi$ are the eigenvalues of $C^{\text{adv}}$, $n$ and $m$ are the number of eigenvalue. Since $L^{\text{diff}}$ is positive semi-definite, all eigenvalues are all real number. The advection term $C^{\text{adv}}$ is antisymmetric, so the eigenvalues are all conjugate pure imaginary or 0. We further project the density $\rho$ through $U$ and get dynamics in the projected space as:

$$\dot{\hat{\boldsymbol{\rho}}} = -U^* C^{\text{adv}} U \hat{\boldsymbol{\rho}} + U^* L^{\text{diff}} U \hat{\boldsymbol{\rho}} = (\text{Diag}(\eta) - \text{j}\text{Diag}(\xi))\hat{\boldsymbol{\rho}}, \tag{12}$$

where $\hat{\boldsymbol{\rho}} = U\boldsymbol{\rho}$. The solution for the ODE equation 12 is:

$$\hat{\boldsymbol{\rho}} = e^{Diag(\eta) - Diag(\text{j}\xi)} \hat{\boldsymbol{\rho}}(0), \tag{13}$$

In traffic flow, $L^{\text{diff}}$ and $C^{\text{adv}}$ are usually not commute, leading them generally not simultaneously diagonalizable in finite dimensions, however, Koopman lifting enables us to approximate their joint evolution in a higher-dimensional linear space, where eigen-decomposition is no longer strictly required.

## 4.2 MACRO DYNAMICS WITH VEHICLE INTERACTION

The dynamics of $\hat{\rho}$ in equation 12 share the same structure as the linear dynamics in the observation space in Koopman theory, which inspires us to lift the macro dynamics to a linear space.

At each time-step, the edge weights $\mathcal{W}$ consists of two matrices $W^{\text{adv}}$ and $W^{\text{diff}}$, which are usually unknown directly from measurements. To reflect how vehicle interactions affect flow dynamics, we use two GNNs to get the edge weights $\mathcal{W} = \{W^{\text{diff}}, W^{\text{adv}}\}$. The two GNN takes $(\mathcal{V}, \mathcal{E})$ as inputs, and outputs the diffusion and advection weights respectively.

To reflect their different physical properties, we design specific structure for diffusion operator and advection operator to ensure $L^{\text{diff}}$ PSD and $C^{\text{adv}}$ antisymmetric. For diffusion operator, we initialize the diffusion edge undirected and adopt Softplus activation for $W^{\text{diff}}$ to keep PSD for diffusion operator. While in advection operator, we reconstruct the graph into a directed graph with edges aligned with the direction of speed field, after obtaining the weights $W^{\text{adv}}$ via advection network, we add a reverse edge of equal weight for each original edge to ensure the antisymmetric. Therefore the corresponding diffusion operator $L_{\text{diff}}$ is symmetric and the advection operator $C_{\text{adv}}$ is antisymmetric.

Inspired by the linear dynamics in the projected space $\hat{\rho}$, we lift the graph features to an observation space $Z_t$ by the Koopman Operator theory.

$$\textbf{Encoder}: Z_t = \phi_Z(\mathcal{G}_t), \tag{14}$$

$$\textbf{Evolver}: Z_{t+1} = K_Z Z_t, \tag{15}$$

$$\textbf{Decoder}: \rho_{t+1} = \psi_Z(Z_{t+1}), \tag{16}$$

where $\phi_Z$ is a GNN to lift the graph features to the observation space $Z$, $\psi_Z$ is an MLP to decode $Z$ to density, and $K_Z$ are a learnable matrix that represents the linear dynamics in the observation space.

We design the loss function for the encoder and decoder as:

$$\mathcal{L}^{\text{macro}}_{\text{encode}} = ||\phi_Z(\mathcal{G}_{t+1}) - K_Z \phi_Z(\mathcal{G}_t)||^2_2, \quad L^{\text{macro}}_{\text{decode}} = ||\bar{\rho}_{t+1} - \rho_{t+1}||^2_1 \tag{17}$$

where $\bar{\rho}$ is the ground truth density as label when training.

In practical traffic flow, the diffusion operator $L^{\text{diff}}$ and advection operator $C^{\text{adv}}$ generally do not commute. We therefore do not assume exact commutation, instead, we penalize the commutator:

$$\mathcal{L}_{\text{JAD}} = ||L^{\text{diff}} C^{\text{adv}} - C^{\text{adv}} L^{\text{diff}}||^2_F \tag{18}$$

where the subscript $F$ is Frobenius norm. This design reduces basis rotation and improves operator-splitting stability under the Lie-Trotter scheme:

$$e^{\Delta t(L^{\text{diff}} - C^{\text{adv}})} \approx e^{\Delta t L^{\text{diff}}} e^{-\Delta t C^{\text{adv}}} \tag{19}$$

this regularizer is fully differentiable and requires no eigen-decomposition thus being numerically stable and further motivates aligning the Koopman operator equation 6.

To ensure the consistence between Koopman operator $K$ with diffusion operator $L^{\text{diff}}$ and advection operator $C^{\text{adv}}$, we set $\theta = \frac{1}{\Delta t}\log(K)$ (principal matrix logarithm) and align real $\theta$ with $\lambda(L^{\text{diff}})$ and imaginary $\theta$ with $\omega(C^{\text{adv}})$, formally:

$$\mathcal{L}_{\text{spec}} = \min_{\Pi}(||\text{Re}(\lambda(\theta)) - \Pi\lambda_L||^2_2 + ||\text{Im}(\lambda(\theta)) + \Pi\omega(\lambda_C)||^2_2) \tag{20}$$

$\Pi$ is permutation operator, we compute $\log(K)$ via a numerically stable real Schur form with small Tikhonov regularization on near unit eigenvalues. When K is not diagonalisable, the Schur-log remains well-defined and differentiable almost everywhere, avoiding branch ambiguities. This explicitly couples the Koopman dynamics to the learned graph-PDE operators, consistent with the splitting used in training.

## 4.3 MICRO DYNAMICS WITH FLOW PROPAGATION

Similar to macroscopic traffic dynamics, vehicle behaviors are also highly nonlinear. Besides, when drivers make decisions, they often consider not only the motion state of themselves, but also surrounding traffic. Therefore, the macroscopic flow propagation also have a direct impact on microscopic dynamics. To avoid the analysis and computation complexity, we also lift the original measurement of vehicle state $x^e_t$ to a higher linear observation space. To reflect how flow propagation affects microscopic vehicle behaviors, we adopt the Koopman Control Theory and design a

CrossAttention block to take the role of Actuation Operator that projects macroscopic influence into the vehicle observation space:

$$\textbf{Encoder}: z_t = \phi_z(x_t^e) \tag{21}$$

$$\textbf{Evolver}: z_{t+1} = K_z z_t + B_z u_t, \quad u_t = \text{CA}(z_t, Z_t) \tag{22}$$

$$\textbf{Decoder}: p_{t+1}^e = \psi_z(z_{t+1}) \tag{23}$$

where CA is CrossAttention block fusing $z_t$ from micro and $Z_t$ from macro, $\phi_z$,$\psi_z$ are MLP, $K_z$ and $B_z$ are trainable matrix, $p_{t+1}^e$ is predicted trajectory. We choose a CrossAttention module rather than a simple linear projection for $u_t$ because traffic influence is highly context-dependent: vehicles in different positions or modes contribute unequally. Attention naturally captures this heterogeneity, aligning with the actuation operator role. To ensure the design of Koopman control is input-state stable (ISS), we need to constrain the output of CrossAttention $u_t$ as bounded and the spectrum radius $\kappa(K_z) < 1$ which ensures that errors decay geometrically with rate $\kappa(K_z)$ and that external influences remain bounded. This provides a formal guarantee that MMCKM will not suffer unbounded error growth over iterative Koopman applications. The detailed proof is provided in Appendix.A.2.

In practical driving process, the microscopic dynamics always present varying features in different driving scenarios: free flow, car-following, lane changing, merging, and emergency maneuvers. Driver intent is commonly modeled as discrete variable and can switch abruptly, implying that a single operator must account for different dynamical regimes. However, directly increasing the dimension of Koopman operator matrix $K_z$ is computationally prohibitive when we calculate the eigenvalue decomposition. Furthermore, different modes may present distinct Koopman spectra and control response. For example, in free flow, drivers tend to keep a constant maximum speed. The spectrum radius $\kappa(K_z)$ is approximate one. The imaginary part of the eigenvalues is small since the oscillation in trajectory present a low frequency variance. Besides, since there is few vehicles, its interaction with surrounding vehicles is weak, and the injects from Cross Attention is small. However, in the lane-changing scenario, vehicles have longitudinal and lateral interaction, leading the coupled oscillation at both real and imaginary eigenvalues of Koopman operator.

To capture various driving scenarios with a low computation cost, we construct a family of Koopman operators consists of multiple $2 \times 2$ complex-valued blocks and diagonal real-valued blocks with distinct initialization schemes such that each operator presents a distinct driving mode. The variation among these operators is achieved by (i) imposing different bounds on the spectral radius of $K_z$ reflecting stability margin, (ii) tuning the coefficients of the complex block control terms $\theta$ to adjust oscillation frequency, and (iii) constraining the maximum actuation operator $B_{\max}$ for actuation strength. The detailed settings and their physical interpretations are provided in Appendix A.2. To determine which operator is most consistent with current traffic environment, we design an Intent Discriminator, which is implemented as a mixture-of-experts (MoE) that evaluates the current vehicle state $x_t^e$ and the graph-based observation $Z_t$ to accordingly select the most plausible Koopman operator from the candidate set. We implement this Intent Discriminator by a MLP through supervised learning, training label are generated in data preprocessing calibrated by acceleration and lane variance. In this way, the high computation burden of a single, over-generalized Koopman operator is alleviated. Instead, the system leverages a structured ensemble of specialized operators, with the Intent Discriminator serving as the gating mechanism that adaptively aligns operator selection with the underlying driving intent.

For the loss function design, the loss functions for the micro encode and decode are calculated by:

$$\mathcal{L}_{\text{encode}}^{\text{micro}} = ||\phi_z(x_{t+1}^e) - K_z \phi_z(x_t^e)||_2^2, \quad L_{\text{decode}}^{\text{micro}} = ||\hat{y}_{t+1}^e - y_{t+1}^e||_2^2 \tag{24}$$

Finally, after training, we only obtain Koopman encoder (GNN and MLP), Koopman operator (matrix) and decoder (MLP) without any eigendecomposition. The computation cost focuses on GNN forward passes and linear Koopman operator iteration.

## 5 EXPERIMENTS

### 5.1 DATASETS

We use trajectory prediction and traffic flow prediction to validate the proposed method. We utilize two highway datasets for our experiments: NGSIM and HighD. We use NGSIM US-101 highway

Table 1: RMSE on NGSIM Dataset. Trajectory error are reported as RMSE. The operator interval in our model is set to 0.1s and 1.0s

| Prediction Horizon (sec) | With historical data | | | Without historical data | | |
|---|---|---|---|---|---|---|
| | BAT | MS-STGCN | Vit-Traj | CV | Ours 1.0sec | Ours 0.1sec |
| 1 | 0.27 | 0.42 | 0.39 | 0.64 | 0.54 | 0.33 |
| 2 | 0.90 | 1.00 | 0.95 | 1.48 | 0.98 | 0.92 |
| 3 | 1.43 | 1.66 | 1.58 | 2.63 | 1.57 | 1.63 |
| 4 | 2.76 | 2.44 | 2.22 | 4.33 | 2.26 | 3.17 |
| 5 | 3.80 | 3.05 | 2.89 | 5.62 | 2.93 | 4.65 |

Table 2: Comparison of accuracy of Different Operator Interval on the HighD dataset

| Interval | 0.04s | 0.1s | 0.2s | 0.4s(*) | 1s |
|---|---|---|---|---|---|
| ADE | 2.84 | 2.06 | 1.88 | 1.65 | 2.90 |

subset, which captures vehicle trajectories sampled at 10 Hz. The HighD datsaset is collected in German highways with a frequency at 25 Hz.

## 5.2 EXPERIMENT SETTINGS

We establish a perception space centered on the ego vehicle, covering three lanes (the ego vehicle lane and two adjacent lanes) over a longitudinal range of 90 m ahead and 60 m behind. The k-NN algorithm is set to identify the 6 nearest vehicles within the perception space to build edge connections. We use kernel density estimation to get density from vehicle positions. We use Gaussian Kernel and set the kernel bandwidth to 25 m. We set the dimension of the observation space $Z$ and $z$ both as 128. For the Intent Discriminator, we set five driving modes: free flow, car-following, lane changing, merging, and emergency. Intent labels are derived directly from raw trajectories using simple, deterministic rules based on longitudinal acceleration, relative headway, and lateral displacement. We assign free flow when vehicles move with stable high speed and minimal interaction; car-following when longitudinal motion is governed by a leading vehicle with reduced headway; lane changing when sustained lateral motion accompanies a lane transition; merging when lateral entry is initiated from on-ramp; and emergency when abrupt braking or rapid acceleration indicates evasive maneuvers. These labels are fully reproducible and require no learned classifier or manual annotation.

We compare against widely-cited recent baselines that are representative of history-dependent predictors: BAT (Liao et al., 2024), MS-STGCN (Tang et al., 2023), Vit-Traj (Cheng et al., 2025). Because history-dependent methods assume access to 3 8 s of trajectories while ours is strictly history-free, our comparison focuses on identical prediction horizons and the same current-state inputs, thereby take CV (Mercat et al., 2019) as a history-free reference. Where intervals differ across papers, we adopt each method's native setting and report our model at matching horizons to avoid conflating sampling effects with modeling capacity. We view these results as conservative for our approach, since removing historical inputs typically disfavors accuracy.

To our knowledge, no prior work evaluates vehicle-centric graph discretization of LWR with diffusion on highway data with publicly-standardized density labels. We therefore treat KDE-estimated density as operational ground truth and report absolute errors to KDE as an internal consistency metric. The default bandwidth of KDE kernel is 25 meters. KDE provides a consistent operational definition, though we acknowledge its limitations and encourage future benchmarks with sensor-derived density labels. Through ablation study, we identify the efficiency of diffusion operator. Cross-paper macro density SOTA comparison are left for future benchmarks.

## 5.3 RESULTS

Table.1 presents prediction accuracy on the NGSIM dataset. We see that the proposed method has a lower prediction error than the SOTA CV method for all prediction horizons. Remarkably, our approach, as a completely history-free model, achieves performance comparable to state-of-the-art methods that require historical trajectory data during the past 3-8 seconds. Traditional sequence-

based methods like BAT and MS-STGCN impose substantial computational overhead through trajectory tracking and storage, limiting real-time applicability. Our Markovian approach eliminates these requirements and also maintains competitive accuracy.

From Table 1, we observe that the operator interval plays a central role in the error accumulation pattern. Our method exhibits an approximately linear growth of error with respect to the number of iterative steps, which is governed by the Koopman sampling interval. The 0.1s operator achieves excellent short-term accuracy (RMSE=0.33 @ 1s) approaching history-dependent models. Nevertheless, its error accumulates more rapidly over longer horizons because each prediction requires repeated applications of the operator. For example, forecasting 5 seconds ahead involves 50 iterations. In contrast, the 1.0 s operator begins with a higher short-term error but demonstrates superior long-term prediction accuracy. For the prediction over 5 seconds, only five iterations are required. This controlled growth in error stands in clear contrast to the exponential error amplification often observed in recurrent architectures. These results suggest that, over extended horizons, our approach can surpass existing algorithms as the dependence on initial historical data diminishes.

To further examine the effect of operator intervals, we use the HighD dataset, which has a higher sampling frequency and provides a more suitable data for evaluating how interval settings affect prediction. We evaluate prediction accuracy for operator intervals at 0.04s, 0.1s, 0.2s, 0.4s, and 1.0s. We compare the Average Displacement Error (ADE) in Table. 2. There is a trade-off between the ability to capture high-frequency and numerical stability: small intervals preserve high-frequency dynamics but suffer numerical instability, while large intervals ensure stability but sacrifice dynamic fidelity. An excessively small interval (e.g., 0.04s) forces each operator to represent minimal dynamic changes, causing eigenvalues to cluster near unity and creating a numerically ill-conditioned system highly sensitive to noise. This configuration requires numerous iterations for prediction, amplifying computational errors. On the other side, an overly large interval (e.g., 1.0s) cannot capture high-frequency dynamics because within a single second, significant portions of driving maneuvers occur compared to complete steering actions typically span 5-6 seconds. The optimal interval of 0.4s achieves the best ADE performance, balancing dynamic representation with numerical stability.

## 5.4 ABLATION STUDY

**Component-wise Analysis** To validate our bidirectional coupling mechanism, we systematically ablate the Intent Discriminator and CrossAttention modules. Table 3 presents RMSE comparisons for four variants: the complete model (MMCKM), Intent Discriminator ablation (MMCKM-I), Koopman control ablation (MMCKM-C), and both components removed (MMCKM-IC).

The results demonstrate complementary roles in multi-scale coupling: The Intent Discriminator primarily enhances short-term predictions (29% improvement at 1s), effectively selecting appropriate dynamics for immediate vehicle behaviors. However, its effectiveness diminishes over longer horizons as the latent macroscopic state $Z_t$ evolves independently, accumulating errors that degrade intent classification accuracy. Maintaining accurate Intent Discrimination would require simultaneous state updates for all surrounding vehicles leading a computationally prohibitive requirement that would negate our method's efficiency advantages.

The Koopman control module proves crucial for long-term stability, reducing error by 37% at 5 seconds compared to MMCKM-C. By injecting macroscopic flow information as control inputs, it maintains the bidirectional coupling essential for accurate long-horizon prediction. Removing this module reduces the system to a single-operator framework, eliminating the critical macro-to-micro influence that constrains vehicle trajectories within physically plausible flow patterns.

**Validation of Diffusion Term** Comparing the complete advection-diffusion model (LC) with advection-only variant (C) on NGSIM. From Table.4, the dramatic degradation confirms our theoretical framework: the diffusion operator $L^{\text{diff}}$ captures how microscopic vehicle perturbations propagate through traffic flow, a phenomenon completely absent in traditional LWR models. Without this term, the model reverts to deterministic flow evolution, unable to represent the stochastic perturbations that characterize real traffic. This result validates our fundamental contribution: the first successful incorporation of microscopic stochasticity into macroscopic PDE models through vehicle-centric discretization.

Table 3: Ablation Study of Different Components on HighD, Operator Interval at 0.2s

| Model | 1s | 2s | 3s | 4s | 5s |
|-------|------|------|------|------|------|
| MMCKM | 0.29 | 0.60 | 1.21 | 1.72 | 2.73 |
| MMCKM-I | 0.74 | 1.39 | 1.96 | 2.90 | 3.81 |
| MMCKM-C | 0.41 | 1.01 | 1.89 | 2.50 | 3.46 |
| MMCKM-IC | 0.80 | 1.74 | 2.54 | 3.48 | 4.62 |

Table 4: Comparison of Removal Diffusion Term on NGSIM

| Model | 1s | 2s | 3s | 4s | 5s |
|-------|------|------|-------|-------|-------|
| LC | 3.2% | 4.1% | 5.6% | 7.4% | 9.5% |
| C | 6.1% | 7.2% | 10.7% | 12.0% | 14.1% |

Table 5: Comparison of Different KDE Bandwidth on NGSIM

| Bandwidth | 1s | 2s | 3s | 4s | 5s |
|-----------|------|------|-------|-------|-------|
| 10 | 6.4% | 7.7% | 11.6% | 12.8% | 14.0% |
| 20 | 4.7% | 5.9% | 7.3% | 9.6% | 11.1% |
| 25 | 3.2% | 4.1% | 5.6% | 7.4% | 9.5% |
| 30 | 4.0% | 5.2% | 7.0% | 9.2% | 10.4% |
| 40 | 4.8% | 6.3% | 8.0% | 10.5% | 11.7% |
| 50 | 5.5% | 6.6% | 8.5% | 11.1% | 12.6% |

**Analysis of KDE Bandwidth Sensitivity** The KDE bandwidth implicitly determines the spatial frequency content of the density field, and thus affects the supervision signal for learning diffusion and advection operators. We compare the performance of macro module with bandwidth $h \in \{10\,\text{m}, 20\,\text{m}, 25\,\text{m}, 30\,\text{m}, 40\,\text{m}, 50\,\text{m}\}$ and report the macro-level RMSE at Table. 5. The bandwidth at 25 meters achieves the best accuracy, while both extremely small and excessively large bandwidths lead to a lower prediction accuracy. By comparing the advection-only variant (C) model in Table. 4 and the complete model (LC) with different kernel length values in Table. 5, we see that the proposed model achieves more accurate under a wide range of bandwidth values, from 20 m to 50 m. We also note that with a extreme small bandwidth of 10 m, LC has a worse prediction than the C model. The possible reason is explained as follows. When the KDE bandwidth is reduced to 10 meters, the density labels become dominated by high-frequency noise, causing arbitrary vehicle perturbation to variance in diffusion term. In other words, according to Eq. 32, small KDE bandwidth brings noise on the density prediction. To fit this noisy, the learned $W_{\text{diff}}$ amplifies the noise rather than capturing genuine perturbation propagation. With the perturbation of $W_{\text{diff}}$, the commutator of $\mathcal{L}_{\text{JAD}}$ and spectral alignment of $\mathcal{L}_{\text{spec}}$ is disrupted, leading the macro evolver $K_Z$ to a wrong direction. As a result, LC with bandwidth 10 performs even worse than C. This result reveals an important insight: diffusion improves macro prediction only when the density supervision carries physically meaningful gradients; otherwise, diffusion becomes a harmful channel that injects noise into operator learning.

## 6 CONCLUSION

We introduce Micro-Macro Coupled Koopman Modeling for traffic prediction, unifying microscopic vehicle dynamics and macroscopic flow evolution within a single Koopman-based architecture. On macro side, we discretize advection-diffusion traffic flow PDE onto vehicle-centric dynamic graph and construct diffusion operator and advection operator ensuring the physical property on graphs. On micro side, we design scenario-adaptive Koopman evolvers selected by Intent Discriminator and utilize Koopman control to inject macro-flow influence into vehicle dynamics. Experiments and ablation studies validate the roles of operator interval, intent gating, KDE kernel sensitivity and macro-to-micro control. Future work will leverage learned edge weights for interpretable interaction measures in vehicle planning and control. Another direction is to expand the framework into urban scenarios under heterogeneous graph structure.

## 7   ETHICS STATEMENT

This research does not involve human subjects, personally identifiable information, or sensitive data. All experiments were conducted on publicly available datasets (HighD, NGSIM), which are widely used in the transportation research community. No personally identifiable information is included in these datasets. The potential societal impact of this work lies in its application to intelligent transportation systems, where improved modeling and prediction could enhance traffic safety and efficiency. At the same time, we acknowledge possible risks, such as misuse in surveillance or decision-making systems that may raise fairness concerns if applied without proper consideration. We encourage responsible and transparent deployment of the proposed methods.

## 8   REPRODUCIBILITY STATEMENT

We are committed to ensuring the reproducibility of our work. To this end, we provide detailed descriptions of our model architecture, training procedures, and evaluation metrics in the main paper and appendix. The datasets used in this study (HighD, NGSIM) are publicly available. We will release our code, including data preprocessing pipelines and training scripts, in an anonymous repository upon publication. Hyperparameters and implementation details are also provided in the supplementary materials to facilitate replication of our results.

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

# A APPENDIX

## A.1 TRAFFIC FLOW EVOLUTION ON GRAPH

Starting from the 2D LWR equation with diffusion on a multi-lane highway in equation 9, where density $\rho$ is described in a Eulerian coordinate $x - y$, and decompose the flux into advection and diffusion components:

$$\frac{\partial \rho}{\partial t} + \nabla \cdot \mathbf{Q} = 0 \tag{25}$$

To avoid averaging vehicle perturbation within cell segmented discretization, instead of discretizing on a fixed Eulerian grid, we adopt a Lagrangian perspective in a dynamical directed weighted graph $\mathcal{G}_t = (\mathcal{V}_t, \mathcal{E}_t, \mathcal{W}_t)$ where vehicles form nodes. At time $t$, we have nodes $\mathcal{V}_t = \{v_1, ..., v_N\}$ representing $N$ vehicles perceived within in range $r_{\max}$ containing node positions $\mathbf{x}_i = (x_i, y_i)$ in 2D space, while density at nodes $\rho_i(t)$ representing local traffic density around vehicle $i$. The vehicles are connected using k-NN to form edges $\mathcal{E}_t$ based on Euclidean distance. Each edge $e_{ij}$ from node $i$ to $j$ defines a direction vector $\mathbf{d}_{ij} = \frac{\mathbf{x}_j - \mathbf{x}_i}{||\mathbf{x}_j - \mathbf{x}_j||}$ and decomposed diffusion weight $w_{ij}^{\text{diff}}$ and advection weight $w_{ij}^{\text{adv}}$ to be determined based on vehicle relative states $w_{ij} = f(v_i, v_j)$. These edges form a non-orthogonal, redundant directional template for approximate gradients. Unlike a regular grid with two orthogonal directions ($x$ and $y$), we have $k$ directions per node that sample the 2D space irregularly and totally $M$ edges.

In a continuous space, a difference is essentially the functional difference between adjacent points, while the gradient is the spatial derivative of the function:

$$\Delta f(x) = f(x + \Delta x) - f(x), \quad \nabla f(x) \approx \frac{f(x + \Delta x) - f(x)}{\Delta x} \tag{26}$$

While on graph, there is no regular grid $\Delta x$ and density $\rho$ is discretized, but exists the adjacent relationship connected by $e_{ij}$. Assume two adjacent nodes $i$ and $j$ carry a scale field $\rho_i$ and $\rho_j$ respectively, the whole density on discretized nodes on graph is collected by a vector $\boldsymbol{\rho} = [\rho_1, ..., \rho_N]$ where $N$ is the vehicle number which depicts the density scalar at the position of each vehicle, the difference alongside edge $e_{ij}$ equals:

$$(\nabla_{\mathcal{G}} \rho)_{e_{ij}} = \rho_j - \rho_i \tag{27}$$

the relationship can be denoted as incident matrix $B$:

$$(B\rho)_{e_{ij}} = -\rho_i + \rho_j = \rho_j - \rho_i \tag{28}$$

Thereby, the gradient on graph can be written as:

$$\nabla_{\mathcal{G}} \boldsymbol{\rho} = B\boldsymbol{\rho} \tag{29}$$

Similarly, the divergence on the graph computes computes the net flux on a node across all incident edges:

$$\nabla \cdot \mathbf{Q} = \text{div}_{\mathcal{G}} \mathbf{Q} = B^\top \mathbf{Q} \tag{30}$$

We take the total flux $\mathbf{Q}$ as a summarization of advection flux $Q^{\text{adv}} + \nabla \cdot \mathbf{Q}^{\text{diff}}$ and diffusion flux, where advection flux $\mathbf{Q}^{\text{adv}} = \boldsymbol{\rho} \mathbf{v}$ transported by velocity field $\mathbf{v}$ and diffusion flux $\mathbf{Q}^{\text{diff}} = -D\nabla\boldsymbol{\rho}$ follows Fick's law for density gradients, $D$ is the diffusion coefficient.

**Diffusion Term** According to Fick's Law, the diffusion flux $\mathbf{Q}^{\text{diff}} = -D\nabla\boldsymbol{\rho}$ can be discretized on graph according equation 29, and consider diffusion coefficient $D$ is anisotropy but related to adjacent dynamical node features, we can derive:

$$\mathbf{Q}^{\text{diff}} = -W^{\text{diff}} B\boldsymbol{\rho}, \quad W_{ij}^{\text{diff}} = f^{\text{diff}}(v_i, v_j) \tag{31}$$

Finally, we calculate the divergence according to equation 30:

$$\dot{\rho}^{\text{diff}} = -\nabla \cdot \mathbf{Q}^{\text{diff}} = -\text{div}_{\mathcal{G}} \mathbf{Q}^{\text{diff}} = B^\top W^{\text{diff}} B\boldsymbol{\rho} := L_{\text{diff}} \boldsymbol{\rho} \tag{32}$$

According to graph Laplacian definition $L = BWB^\top$, this equation can be described through a diffusion operator $L_{\text{diff}}$, which is not a graph Laplacian but called edge Laplacian:

$$\dot{\boldsymbol{\rho}}^{\text{diff}} := L_{\text{diff}} \boldsymbol{\rho}, \quad L_{\text{diff}} = B^\top W^{\text{diff}} B \tag{33}$$

In diffusion term which describes the stochastic perturbation propagates along density gradient, it's an entropy production process, leading the dissipation entropy nonnegative:

$$\frac{d}{dt}\mathcal{E}(\boldsymbol{\rho}) = \frac{d}{dt}(\frac{1}{2}||\boldsymbol{\rho}||_2^2) = \boldsymbol{\rho}^\top\dot{\boldsymbol{\rho}} = \boldsymbol{\rho}^\top(B^\top W^{\text{diff}}B)\boldsymbol{\rho} = (B\boldsymbol{\rho})^\top W^{\text{diff}}(B\boldsymbol{\rho}) \geq 0 \qquad (34)$$

define $y = B\boldsymbol{\rho}$, this equation turns to:

$$y^\top W^{\text{diff}}y \geq 0 \qquad (35)$$

thereby, $W^{\text{diff}}$ must be positive seimi-definite (PSD).

**Advection Term** In continuous space, $\mathbf{Q}^{\text{adv}} = \rho(\mathbf{x}, t) \cdot \mathbf{v}(\mathbf{x}, t)$, which is a vector field, represents density transported by speed field. On graph, there is no more continuous speed field $\mathbf{v}(x)$, but discretized nodes and edges. In this setting, the direction of speed field $\mathbf{v}(x)$ is not solely 2D $x$ and $y$ coordinates, but formed by $M$ non-orthogonal, redundant direction vector $\mathbf{d}_{ij} = \frac{\mathbf{x}_j - \mathbf{x}_i}{||\mathbf{x}_j - \mathbf{x}_i||}$.

In continuous condition, the flux via a section $A$ is:

$$\mathbf{Q}^{\text{adv}} = \int \rho\mathbf{v} \cdot \mathbf{n} \cdot dA \qquad (36)$$

where $\mathbf{n}$ is the normal vector of section $A$. While in graph, there is no more section vertical to speed field but connected with edge channels. We account the flux passing the edge between node $i$ and $j$. The flux on edge is bidirectional $\mathbf{Q}_{ij} = (\mathbf{Q}^{\text{adv}}_{i\to j} - \mathbf{Q}^{\text{adv}}_{j\to i})$, and the flux on edge for one direction equals:

$$\mathbf{Q}^{\text{adv}}_{i\to j} = \rho_i \cdot (\mathbf{v}_i \cdot \mathbf{d}_{ij}) \cdot dA_{ij}$$
$$\mathbf{Q}^{\text{adv}}_{j\to i} = \rho_j \cdot (\mathbf{v}_j \cdot \mathbf{d}_{ji}) \cdot dA_{ji} \qquad (37)$$

define:

$$a_{ij} := (\mathbf{v}_i \cdot \mathbf{d}_{ij}) \cdot dA_{ij}, \quad a_{ji} := -(\mathbf{v}_j \cdot \mathbf{d}_{ij}) \cdot dA_{ji} \qquad (38)$$

we decompose $\rho_i$ and $\rho_j$ to average and difference:

$$\rho_i = \bar{\rho} + \frac{1}{2}(\rho_i - \rho_j), \quad \rho_j = \bar{\rho} - \frac{1}{2}(\rho_i - \rho_j), \quad \bar{\rho} = \frac{1}{2}(\rho_i + \rho_j) \qquad (39)$$

then, the advection flux on edge is converted to:

$$\mathbf{Q}_{ij} = \mathbf{Q}^{\text{adv}}_{i\to j} - \mathbf{Q}^{\text{adv}}_{j\to i} = a_{ij}\rho_i - a_{ji}\rho_j = \frac{1}{2}(\rho_i - \rho_j)(a_{ij} + a_{ji}) + \bar{\rho}(a_{ij} - a_{ji}) \qquad (40)$$

the first term can be rewritten on graph as $W^{\text{adv}}B\boldsymbol{\rho}$ and $W^{\text{adv}} = \frac{1}{2}(a_{ij} + a_{ji})$.

According to equation 25, we calculate the divergence on node:

$$\dot{\boldsymbol{\rho}}^{\text{adv}} = -\nabla \cdot \mathbf{Q}^{\text{adv}} = -\text{div}_{\mathcal{G}}\mathbf{Q}^{\text{adv}} = -B^\top\mathbf{Q}^{\text{adv}} \qquad (41)$$

and the discrete conservation law requires the total density is preserved:

$$\mathbf{1}^\top\dot{\boldsymbol{\rho}}^{\text{adv}} = -\mathbf{1}^\top B^\top\mathbf{Q}^{\text{adv}} = 0 \qquad (42)$$

since $B\mathbf{1} = 0$, this is guaranteed if $\mathbf{Q}^{\text{adv}}$ depends only on the difference $B\boldsymbol{\rho}$ and the second term $\bar{\boldsymbol{\rho}}(a_{ij} - a_{ji})$ vanished.

Finally, the equation 41 can written as:

$$\dot{\boldsymbol{\rho}}^{\text{adv}} = -B^\top W^{\text{adv}}B\boldsymbol{\rho} \qquad (43)$$

use a Laplacian operator $C^{\text{adv}}$ to denote:

$$\dot{\rho}^{\text{adv}} := -C^{\text{adv}}\rho, \quad C^{\text{adv}} = B^\top W^{\text{adv}}B \qquad (44)$$

Furthermore, according to physical law, we require advection is energy-preserving because it's non-dissipative:

$$\frac{d}{dt}(\frac{1}{2}||\boldsymbol{\rho}||_2^2) = \boldsymbol{\rho}^\top\dot{\boldsymbol{\rho}} = \boldsymbol{\rho}^\top C^{\text{adv}}\boldsymbol{\rho} = 0 \qquad (45)$$

this requires $C^{\text{adv}}$ to be an antisymmetric matrix $C^{\text{adv}\top} = -C^{\text{adv}}$. To ensure $C^{\text{adv}}$, we develop a special parameterization method built from original graph.

We first reconstruct the graph into a directed graph with edges aligned with the direction of speed field, and let $A^{\text{line}} \in \{0,1\}^{M \times M}$ denote the line-graph adjacency, where $A^{\text{line}}[e, e'] = 1$ iff edges $e$ and $e'$ share a node. Define a symmetric locality mask:

$$M^{\text{loc}} := \frac{1}{2}(A^{\text{line}} + A^{\text{line}\top} - I) \tag{46}$$

and introduce an unconstrained parameter matrix $P \in \mathbb{R}^{M \times M}$. We parameterize the advection weights and operator by:

$$W^{\text{adv}} := M^{\text{loc}} \circ (P - P^\top), \quad C_{\text{adv}} := B^\top W_{\text{adv}} B \tag{47}$$

where $\circ$ is the Hadamard product and $B$ is incident matrix. Because $M^{\text{loc}}$ is symmetric and $P - P^\top$ is skew-symmetric, we have:

$$(W^{\text{adv}})^\top = M^{\text{loc}} \circ (P^\top - P) = -W^{\text{adv}} \tag{48}$$

hence, we can get:

$$C^{\text{adv}\top} = (B^\top W^{\text{adv}} B) = B^\top (W^{\text{adv}})^\top B = -C^{\text{adv}} \tag{49}$$

therefore $C^{\text{adv}}$ is antisymmetric, ensuring non-dissipative advection term.

In summary, equation 25 can be discretized on graph:

$$\dot{\boldsymbol{\rho}} = -C^{\text{adv}}\boldsymbol{\rho} + L^{\text{diff}}\boldsymbol{\rho} \tag{50}$$

## A.2 KOOPMAN OPERATOR DESIGN

In equation 22, even if $K_z$ is stable, if control $u_t$ is infinite and no constrain on $B_z$, $z_t$ is also divergent. Furthermore, in practice, the augmented Koopman operator $[K_z|B_z]$ of vehicle dynamics is constraint by physical limitations such as limited steering angle, throttle and acceleration. Thereby, we need to ensure $u_t$ is input-state stable (ISS), equals to:

$$\exists\, \text{const } c \geq 1, \lambda \in (0, 1)$$
$$\text{s.t. } ||z_t|| \leq c\lambda^t ||z_0|| + \frac{cB_z}{1 - \lambda} \sup_{0 \leq \tau \leq t-1} ||u_\tau|| \leq c\lambda^t ||z_0|| + \frac{cB_z}{1 - \lambda} U_{\max} \tag{51}$$

To ensure that the observation $z$ is always bounded, we require that the output of CrossAttention $u_t$ is bounded and $|\lambda| < 1$. For the first requirement, we add a Sigmoid operator before the final output of the CrossAttention module.

For the second requirement, it equals to ensure the spectrum radius $\kappa(K_z) < 1$. In our design, the Koopman operator $K_z(\lambda)$ is a trainable approximately diagonal matrix with real and complex eigenvalues. We use $Nc$ $2 \times 2$ complex blocks and $Nr$ $1 \times 1$ real blocks to construct $K_z(\lambda)$, where $Nc$ and $Nr$ is a hyperparameter related to Koopman operator dimension. To ensure $\kappa(K_z) < 1$, we need to simultaneously promise the mode of each complex and real blocks strictly less than 1. For each complex block, we construct it with two learnable parameter, radius $R$ and rotation $\theta$ to parameterize the complex conjugate eigenvalue pairs $\lambda = Re^{(\pm i\theta)}$, apply to $2 \times 2$ blocks:

$$K_c = R \times \begin{bmatrix} \cos(\theta) & -\sin(\theta) \\ \sin(\theta) & \cos(\theta) \end{bmatrix} \tag{52}$$

we further constrain each radius $R$ less than 1 by applying $R = \kappa_{\max} \times \text{sigmoid}(\eta)$, where $\eta$ is the eigenvalue of advection operator $C^{\text{adv}}$ which governs the traffic flow propagation, while $\theta_{\text{mean}}$ and $\theta_{\text{std}}$ represents the vibration frequency as diffusion operator $L^{\text{diff}}$. In real block, we apply same constraint on $R = \kappa_{\max} \times \text{sigmoid}(\eta)$ to ensure stability. Furthermore, through this design, we can directly record $\max(R)$ to quickly estimate spectrum radius of each Koopman operator.

Even if we ensure $u_t$ is bounded by a Sigmoid operator, while training, $B_z$ is possible to divergence to big enough to break the physical constraint, to avoid it, we constrain it by $B_{\max} \times \text{Tanh}(B_z)$.

Table 6: The Hyperparameter of Each Mode

| Mode | $\kappa_{\max}$ | $B_{\max}$ | $\theta_{\text{std}}$ | $\theta_{\text{mean}}$ |
|---|---|---|---|---|
| free flow | 0.95 | 0.20 | 0.01 | 0.00 |
| car-following | 0.85 | 0.60 | 0.02 | 0.00 |
| lane changing | 0.90 | 0.75 | 0.08 | 0.25 |
| merging | 0.88 | 0.80 | 0.05 | -0.15 |
| emergency | 0.70 | 0.40 | 0.01 | 0.00 |

These design is convenient for us to introduce prior knowledge into different Koopman operator because we only need to set different $\kappa_{\max}$, $\theta_{\text{std}}$, $\theta_{\text{mean}}$ and $B_{\max}$ for different driving mode. In free flow mode, persist speed and fixed direction is reflected by the slowly decaying rate, equals to ($\kappa_{\max} \to 1$), small perturbation from stochastic vehicle behavior ($\theta_{\text{std}}, \theta_{\text{mean}} \to 0$), and trivial external influence ($B_{\max} \to 0$), the difference of car-following mainly comes from front vehicle influence which can be reflected by a larger $B_{\max} > 0$. While in lane-changing and merging, the external influence is obvious ($B_{\max} \to 1$) and there is explicit vibration period ($\theta_{\text{mean}} \neq 0$), here we set lane changing initialized to be larger than 0 and merging initialized to be smaller than zero is to avoid lane changing and merging to be trained to same mode quickly. Finally, for emergency mode, vehicle accelerates quickly, represented by $\kappa_{\max} < 1$ while suffered from larger external influence $B_{\max} > 0$.

We list all parameter setting in Table.6

## A.3 RUNTIME EFFICIENCY ANALYSIS

Our Koopman-based formulation provides a clear inference-time advantage over conventional spatiotemporal models. Unlike recurrent or attention-based architectures that must repeatedly unroll temporal dynamics or compute global pairwise attention, our approach decouples temporal evolution into single linear Koopman operator propagations, allowing the full multi-step future to be generated in one pass. Consequently, the inference complexity is dominated by a sparse graph message passing stage with $\mathcal{O}(Ed)$ cost, where $E = k \cdot N$ is the number of edges, plus Koopman evolution whose cost depends solely on the latent dimension $d$ of complexity $\mathcal{O}(Td^2)$, totally $\mathcal{O}(kNd + Td^2)$. Compared to the complexity of spatiotemporal GNN of $\mathcal{O}(T(Nd^2 + kNd))$ and spatiotemporal Transformer $\mathcal{O}(T^2N^2d)$, the computation complexity is greatly reduced.

Experientially, on an NVIDIA 4090, our model achieves an average inference time 0.20 s for 10 batches with a batchsize of 128, which already satisfies the real-time deployment.

## A.4 USE OF LLMS STATEMENT

We used large language models (LLMs) during the process of preparing this manuscript. Specifically, LLMs were employed for language polishing, grammar checking, and improving clarity of expression in certain sections of the paper. All technical content, derivations, experiments, and results were designed, implemented, and verified by the authors. The authors take full responsibility for the correctness and integrity of the scientific contributions presented in this work.

