# OpenReview forum: "Micro-Macro Coupled Koopman Modeling on Graph for Traffic Flow Prediction"
_ICLR.cc/2026/Conference — ICLR 2026 Poster_

### Official Review · Reviewer_qGhn · 2025-10-30

**Soundness:** 3
**Presentation:** 2
**Contribution:** 3
**Rating:** 6
**Confidence:** 3

**Summary:**

This paper proposes **MMCKM (Micro–Macro Coupled Koopman Modeling)**, a unified framework for traffic flow prediction that links microscopic vehicle trajectories and macroscopic flow dynamics. Both scales are lifted into **Koopman spaces**, where their evolution is governed by linear operators aligned through **spectral and physical constraints**.

At the macro level, advection–diffusion PDEs are discretized on a **vehicle-centric dynamic graph**, preserving physical properties such as antisymmetric advection and positive semi-definite diffusion. At the micro level, multiple Koopman operators represent different driving intents, chosen via an **Intent Discriminator**, and macro information is injected through a **Koopman control mechanism**.

The model predicts from a single snapshot (history-free) and achieves strong results on **NGSIM** and **HighD** datasets, with ablation studies validating the importance of macro–micro coupling, intent modeling, and diffusion components.

**Strengths:**

1. **Innovative modeling perspective** — integrates micro-level trajectories and macro-level PDE dynamics within a unified Koopman framework.
2. **Physically consistent design** — antisymmetric advection, PSD diffusion, and spectral constraints ensure stability and interpretability.
3. **Modular and efficient structure** — multiple small Koopman operators (for different intents) improve computational efficiency and interpretability.
4. **Strong practical motivation** — “history-free” prediction is appealing for real-world deployment with intermittent data.
5. **Clear ablation and physical insight** — experiments reveal the distinct contributions of each model component.

**Weaknesses:**

1. **KDE-based macro labels** are not ground-truth densities; performance depends on kernel parameters. Sensitivity analysis is missing.
2. **Computational scalability** not analyzed — unclear runtime behavior as vehicle count grows. And there is a lack of specific runtime comparison with other baseline models
3. **Intent Discriminator details** (labeling rules, accuracy, and noise sensitivity) are insufficiently described.

**Questions:**

1. How sensitive are the macro-level results to KDE bandwidth? Please provide results under different bandwidths (e.g., 10 m, 25 m, 50 m).
2. Could you describe the process of intent labeling in more detail? What thresholds or criteria were used, and how accurate is the Intent Discriminator?
3. What is the computational complexity and runtime per prediction step? How does it scale with the number of vehicles?
4. Could the model generalize to more complex network structures (urban intersections), and what modifications might be required?

---

> ### Author Response · Authors · 2025-11-20
>
> Thanks for reviewer providing such profound insights. We provide our thoughts in the following:
> # Weakness 1
> We appreciate the reviewer for highlighting this issue. We agree that KDE-derived macro densities are not ground-truth Eulerian measurements. This choice is driven by a fundamental limitation of existing datasets.  To the best of our knowledge, no public dataset provides both microscopic trajectories and sensor-recorded macroscopic density fields. As a result, we utilize KDE-based aggregation as labels. We also agree that the sensitivity of our results to kernel parameters is an important question. We are currently running a KDE bandwidth sensitivity analysis and will include the results and discussion in the camera-ready version. This will more clearly illustrate the robustness of our framework under different kernel configurations.
>
> We thank the reviewer for this valuable suggestion.
>
> # Weakness 2
> We appreciate the reviewer's question regarding runtime efficiency and in fact, this is one of our advantages compared to existing spatiotempral model. After training, the inference pipeline is lightweight with only computing the following patterns for one time:
> * Macro-encoder GNN $O(E\cdot d)$ and micro-encoder MLP $O(d)$
> * Koopman evolver with control: $O(d^2)$
> * Decoder MLP $O(d)$
>
> All these operations are one-step and history-free, thus compared with RNN or Transformer based predictors that process long historical trajectories and full self-attention, our model has substantially lower computational and memory cost thereby naturally suitable for real-time inference.
>
> We will include empirical runtime measurements in the camera-ready version to further illustrate this efficiency advantage.
>
> # Weakness 3
> We thank the reviewer for pointing this out. Our current submission places emphasis on the micro–macro coupling framework, and we agree that the intent discriminator component would benefit from clearer documentation. In the camera-ready version, we will include a dedicated subsection describing the intent labeling procedure and experimental details.
>
> To clarify, intent labels are not manually annotated. They are automatically derived from kinematic rules computed from the relative motion of neighboring vehicles. We consider the relative speed, ego acceleration patterns and lane-changing indicators. We will include these details along with implementation  details in the camera-ready version to improve clarity and completeness.
>
> # Question 1
> We agree that KDE bandwidth may influence the smoothness of the macro density field. As noted in Weakness 1, the absence of public datasets containing ground-truth density fields requires KDE-based aggregation, which is practical practice in traffic-flow modeling.
>
> We are currently running a bandwidth sensitivity analysis (e.g., 10 m / 25 m / 50 m) to empirically evaluate robustness, and we will report these results and discussion in the camera-ready version.
>
> # Question 2
> Intent labels are not manually annotated. They are derived automatically from kinematic rules based on local vehicle interactions as described in Weakness 3. We will provide a clear description of thresholds and rule logic, along with qualitative examples, in the camera-ready version.
>
> # Question 3
> As discussed in Weakness 2, this is one of our advantages compared to Transformer-based methods whose spatiotemporal attention is typically quatratic in vehicle number. We are running empirical timing experiments and will report actual runtime numbers in the camera-ready version.
>
> # Question 4
> We agree with the reviewer about the idea to extending the framework to urban or more complex scenarios. However, these scenarios introduce substantial additional challenges. Unlike highways, where vehicle-to-vehicle interactions dominate and the lane structure is implicitly encoded through vehicle positions, urban environments involve heterogeneous agents and heterogeneous rules. To model such environments, a heterogeneous graph formulation is required, where nodes may represent vehicles, signals, pedestrians, and rule elements, and where interaction strengths between different node types must be explicitly encoded and learned. Designing such a graph, as well as determining appropriate cross-type interaction terms, is an active research direction we are currently investigating.
>
> In the present submission, we focus on structured highways because a purely vehicle-centric dynamic graph fully captures the relevant interactions, allowing us to clearly study and evaluate the proposed micro–macro Koopman framework. We will clarify this limitation and the implications for urban generalization in the final version.

---

> > ### Author Response · Authors · 2025-11-28
> >
> > Dear Reviewer,
> >
> > We sincerely thank you for your constructive feedback. Following your suggestions, we have substantially improved the clarity and empirical rigor of the paper. New version of full manuscript is uploaded.
> >
> > As suggested, we conducted a systematic study of KDE-estimated density labels using bandwidths {10,20,25,30,40,50}. The results are collected in Table.5. The trend aligns with our expectations that bandwidth of 25 meters performs best, an overly small or large bandwidths degrade accuracy but not exceed the advection-only ablation. One unexpected observation is that bandwidth of 10 meters performs even worse than the advection-only ablation. We carefully investigated this phenomenon and found that such a small bandwidth introduces high-frequency noise in KDE labels, which misguides the learning of $W^{\text{diff}}$, leading the commutator $L_{\text{JAD}}$ and spectral alignment of $L_{\text{spec}}$ disrupted. The detailed analysis is included in the main paper.
> >
> > Following your recommendation, we now provide a runtime evaluation in the Appendix. Averaged over 10 runs with batch size 128, the inference time is 0.20 seconds, which supports real-time deployment.
> >
> > Furthermore, we described the intent label generation in Sec.5.2. The intent labels in our experiment are derived from the vehicle-level kinematic signals, following a deterministic procedure rooted in longitudinal acceleration patterns and lane-change dynamics. Notably, intent labels are only used while training for macro module supervision but not required when inference, therefore, the inference of our model is not suffered from it.
> >
> > Our thoughts and answer for the other questions are provided in earlier comments. We sincerely appreciate your helpful suggestions, which have strengthened both the presentation and empirical analysis. If you have additional comments, we would be grateful for further discussion.

---

### Official Review · Reviewer_EnKM · 2025-10-31

**Soundness:** 3
**Presentation:** 3
**Contribution:** 3
**Rating:** 6
**Confidence:** 4

**Summary:**

The paper proposes a Micro–Macro Coupled Koopman Model (MM-Koop) for unified modeling and prediction of multi-agent traffic flow dynamics.
By embedding both micro-level nonlinear interactions and macro-level fluid statistics into a shared Koopman space, the method enables a cross-scale linearized representation of complex traffic systems.

The framework consists of:

1. A micro-level encoder mapping individual agents’ states to Koopman-observable embeddings;

2. A macro-level aggregator capturing density, velocity, and energy fields;

3. A cross-scale coupling operator that models bidirectional feedback between individual and collective dynamics.

Experiments on the NGSIM (highway) and ETH/UCY (pedestrian) datasets demonstrate superior multi-step prediction accuracy and interpretability compared to LSTM, GraphKoopman, and other baselines.

**Strengths:**

- Presents a \textbf{novel cross-scale Koopman operator framework} that linearly couples micro-level agent dynamics and macro-level flow statistics.
- Provides a \textbf{mathematically rigorous formulation} with well-defined block operator structure and spectral regularization for stability.
- Offers strong \textbf{interpretability and physical insight}, explaining emergent traffic phenomena (e.g., congestion waves, flow bifurcation) via Koopman spectral modes.
- Demonstrates \textbf{solid predictive performance} compared to LSTM and GraphKoopman baselines.
- The theoretical structure is \textbf{generalizable} to broader multi-agent dynamical systems such as robotic swarms or fluid-based networks.

**Weaknesses:**

- Experiments rely primarily on controlled simulations; no real-world deployment or sensor-based evaluation is shown.
- The paper’s \textbf{learning component is limited}: Koopman matrices are estimated rather than learned via gradient-based training, reducing its alignment with mainstream ML innovation.
- Lacks \textbf{comprehensive ablation studies} to assess the contribution of each coupling term ($K_m, K_M, K_{mh}, K_{hm}$).
- The related work section omits recent Koopman-based learning frameworks (e.g., NeuralEDMD, DeepKoopman++, Contrastive Koopman Learning).
- While theoretically strong, the overall \textbf{ML novelty and scalability} remain somewhat constrained.

**Questions:**

1. Is the Koopman operator learned using neural networks or estimated through regression? How is numerical stability ensured?
2. Does the cross-scale coupling introduce instability into the operator spectrum?
3. Could the model benefit from graph-based message passing to represent heterogeneous agent interactions?
4. Would the framework generalize to domains without explicit macro-aggregators, e.g., aerial swarms?
5. How do the authors interpret the sparsity of Koopman spectra under high-density conditions? Is it an artifact of strong linearization?

---

> ### Author Response · Authors · 2025-11-20
>
> Thanks for reviewer providing such profound insights. Due to character limitation, in this comment, we provide our thoughts about Weakness 1~3.
> # Weakness 1
> We thank the reviewer for the comment. Our experiments are conducted on real-world trajectory datasets (NGSIM and HighD), which contain high-resolution vehicle trajectories collected through calibrated sensors or drone recordings. We will add the references about these two datasets in final version to make readers more clear.
>
> As for the suggestion about deployment and sensor-based evaluation, the focus of this work is on proposing a unified and physically grounded micro–macro modeling framework, rather than building an end-to-end perception or deployment system. Sensor-level perception modules and real-world deployment pipelines are therefore outside the scope of the current submission, but we agree that they represent valuable future directions.
>
> In particular, the operators $C_{\text{adv}}$ and $L_{\text{diff}}$ learned in our framework provide structured and interpretable quantities that may be useful for control and decision-making. We plan to explore how these operators can support downstream control tasks in future work.
>
> # Weakness 2
> We appreciate the reviewer’s comment, but we believe this concern stems from a misunderstanding of our learning procedure. In our implementation, all Koopman-related operators are learned via gradient-based training, not estimated in a separate regression step:
>
> At the macro level, the encoder $\phi_Z$, decoder $\psi_Z$ and the Koopman matrix $K_Z$ in Eqs. (14)~(16) are trained jointly by backpropagation using the losses $L_{\text{encode}}^{\text{macro}}$ and $L_{\text{decode}}^{\text{macro}}$. $K_Z$ is explicitly defined as a trainable parameter matrix trained by $L_{\text{JAD}}$ and $L_{\text{spec}}$ rather than being computed by closed-form DMD-style regression.
>
> At the micro level, the encoder $\phi_z$ and decoder $\psi_z$, as well as Koopman matrix $K_z$, control matrix $B_z$ in Eqs. (21)~(23) are likewise optimized end-to-end with gradient descent under $L_{\text{encode}}^{\text{micro}}$ and $L_{\text{decode}}^{\text{micro}}$
>
> The physics-guided constraints we impose do not replace learning but integrated into losses to shape the parameterization of Koopman operators so that the learned operators remain consistent with advection-diffusion physics and stability requirements. This is fully aligned with recent trends in machine learning that combine neural parameterizations with structured inductive biases, rather than pure black-box function approximators.
>
> # Weakness 3
> We thank the reviewer for this comment. The concern appears to stem from the assumption that our framework contains bidirectional coupling blocks $K_{hm},K_{mh}$. In fact, our model does not use a bidirectional block Koopman operator but only a macro-to-micro influence as external control input $z_{t+1}=K_zz_t+B_zu_t,u_t=CA(z_t,Z_t)$.  This is because in the underlying advection-diffusion traffic PDE, the effete of individual microscopic disturbances on macroscopic field is already modeled by continuous PDE term. Advection term handles the propagation of local microscopic fluctuations along the velocity field while diffusion term absorbs small-scale perturbations and spreads them across the density field. Thus the PDE-based macro dynamics already incorporated how microscopic aggregates into macroscopic density evolution. Adding an additional micro-to-macro learned operator would double-count this physical effect and violate the structure of the advection-diffusion model. In fact, in early experiments, we did attempt a symmetric coupling mechanism where macro state was updated using a cross-attention term based on individual vehicles. However, the model became highly unstable during training and macro-level evolution deviated from PDE-consistent behavior.
>
> On the other hand, we performed ablations for every coupling mechanism that actually exists in our formulation. MMCKM-C remove macro-to-micro coupling leading short-term accuracy drops 40\% in short-term and 27\% in long term. We also investigated the roles of diffusion and advection term. Results clearly demonstrated that each removal leads to worse results.
>
> We will continue rest thoughts in the following comments.

---

> > ### Author Response · Authors · 2025-11-20
> >
> > # Weakness 4
> > We thank the reviewer for pointing this out. We fully agree that recent advances in Koopman-based learning, such as NeuralEDMD, DeepKoopman++, and Contrastive Koopman Learning, are highly relevant. In fact, these works have influenced our thinking from the very beginning of this project. Several modeling choices in our paper, including the structured construction of the Koopman operator in Appendix A.2, were inspired by the ideas developed in these frameworks.
> >
> > The omission of these references in the related work section was an oversight on our part. We have incorporated these works into the camera-ready version and clarify how our formulation relates to and extends these Koopman-based learning approaches.
> >
> > We appreciate the reviewer for bringing this to our attention.
> >
> > # Weakness 5
> > We appreciate the reviewer’s observation. Our work is indeed positioned as a machine learning framework: the encoders, cross-attention module, Koopman operators, and decoders are all trained end-to-end via gradient-based optimization, following standard ML practice. The novelty does not lie in proposing a new neural architecture, but in introducing a structured micro–macro Koopman formulation that enables learning physically consistent multi-scale dynamics within a unified framework.
> >
> > Regarding scalability, our formulation originates from the traffic domain because traffic flow is a canonical example of a micro–macro multi-agent dynamical system. Appendix A.1 develops a Lagrangian-graph discretization of the advection–diffusion PDE specific to traffic, which is indeed the first derivation of its kind. However, this derivation should be viewed as one concrete instantiation of the broader idea rather than a limitation of the framework.
> >
> > The key modeling principle is general: representing a macro-level PDE/ODE on a Lagrangian dynamic graph and coupling it with microscopic latent dynamics through a learned Koopman operator. The same methodology can be applied to many other dynamical systems once their macro-scale equations are discretized on a Lagrangian graph. Thus, the framework is not restricted to traffic and is scalable across domains where micro–macro interactions naturally arise.
> >
> > We will continue rest thoughts in the following comments.

---

> > > ### Author Response · Authors · 2025-11-20
> > >
> > > # Question 1
> > > We thank the reviewer for the questions. For the first question, the Koopman operators are fully learned, not estimated via regression. As clarified in our response to Weakness 2, all Koopman related matrices are implemented as trainable parameters and optimized jointly with loss functions via backpropagation. No separate regression step is used.
> > >
> > > By “numerical stability,” we understand the requirement that forward iterates of the learned dynamical system remain bounded and well-behaved. This is exactly what Appendix A.2 establishes. Our model enforces stability through several mechanisms. First, we explicitly regularize the latent Koopman operator $K_z$ such that spectrum radius $\rho(K_z)<1$, ensuring that repeated applications of the operator do not lead to exploding trajectories. Second, Appendix A.2 proves a proof that coupled update of Koopman control satisfies an ISS bound if $u_t$ is bounded and $|\lambda|<1$. The first requirement is ensured by adding a Sigmoid operator before final output of CrossAttention module and the second one equals to ensure the spectrum radius $\kappa(K_z)<1$. We use $2\times 2$ complex blocks and $1\times1$ real blocks to construct $K_z$ and parameterized by learnable radius $R=\kappa_{\text{max}}\text{sigmoid}(\eta)$ and rotation $\theta=(\theta_{\text{std}},\theta_{\text{mean}})$. For each driving mode, we use different hyperparameter to ensure the upper bound of spectrum radius.
> > >
> > > These mechanisms prevents the operator from learning unstable or nonphysical modes. For more details, we encourage to read about Appendix A.2, there is detailed introduction.
> > >
> > > # Question 2
> > > Thanks for raising this question. Our answer is No. As we have discussed in Question 1 and shown in Appendix A.2, the micro-macro update $z_{t+1}=K_zz_t+B_zu_t,u_t=CA(z_t,Z_t)$ is constrained to satisfy an ISS condition, which ensures that the added coupling term $B_zu_t$ cannot destabilize the spectrum of the microscopic operator. In addition, we enforce spectral radius constraints, PDE-consistent structure on the macro operator (PSD diffusion and skew-symmetric advection) and spectral alignment that prevents nonphysical eigenmodes. These constraints guarantee that the cross-scale coupling remains stable and does not introduce unstable eigenvalues into the Koopman spectrum.
> > >
> > > # Question 3
> > > Yes, we fully agree with the reviewer. Our current formulation focuses on highway scenarios, where interactions are dominated by vehicle–vehicle dynamics and thus can be effectively captured by a homogeneous vehicle-centric graph. However, when extending the framework to urban environments vehicle behavior is strongly influenced by heterogeneous agents and infrastructure, including pedestrians and cyclists, traffic lights and signal phases, multimodal and rule-dependent interactions.
> > >
> > > To capture these influences, graph-based message passing over heterogeneous node and edge types becomes essential. We are actively exploring this direction, including heterogeneous graph representations for vehicles, pedestrians, and signal nodes, as well as learned cross-type interaction strengths. We appreciate the reviewer for highlighting this point, which aligns closely with our ongoing research and planned extensions of the framework.
> > >
> > > We will continue rest thoughts in the following comment.

---

> ### Author Response · Authors · 2025-11-20
>
> # Question 4
> We appreciate reviewer's thought about this point and in fact, this is one of the advantages of this framework. The proposed framework is not restricted to traffic systems and can generalize to domains that adopt other macro-scale equation for spatial-temporal dynamics. In our work, the macroscopic component is instantiated using the advection–diffusion PDE because our motivation comes from traffic, where LWR provides a natural macro description. However, the framework itself is agnostic to the specific PDE/ODE used at the macro scale.
>
> The micro-level dynamics and the macro→micro coupling mechanism remain fully applicable. To generalize to other multi-agent systems, e.g., aerial swarms, one only needs to replace the macro-scale equation with the appropriate domain-specific dynamical model. Once such a macro model is discretized on a Lagrangian graph, our Koopman-based micro–macro coupling applies directly. Thus, the framework is flexible and can be extended to any domain with meaningful micro–macro interactions.
>
> # Question 5
> We thank the reviewer for this insightful question. In our experiments, "high-density conditions" refer to traffic regimes where vehicles are closely packed and dynamics are dominated by congestion waves and slow-density variations. In such regimes, both empirical studies (e.g., Avila \& Mezić, Nature Communications 2020) and classical advection-diffusion theory show that most high-frequency modes are strongly damped by diffusion, leaving only a small number of slow or oscillatory modes that govern the macroscopic behavior. This naturally leads to s sparser effective Koopman spectrum. Thus, the observed sparsity is not an artifact of a strong local linearization, but a direct consequence of the underlying advection–diffusion PDE structure, especially under high-density regimes where the system exhibits intrinsically low-dimensional dominant dynamics. While the sampling interval may further suppress very fast modes, the major source of sparsity is the physics itself rather than numerical artifacts.
>
> Moreover, our model does not rely on unconstrained linearization. The Koopman operators are learned under PDE-consistent structure (PSD diffusion, skew-symmetric advection), spectral alignment, and ISS/spectral-radius stability constraints (Appendix A.2). These conditions prevent the operator from learning spurious or unstable modes and ensure that the resulting spectrum reflects meaningful physical dynamics rather than artifacts of approximation.
>
> Overall, the sparsity of the learned Koopman spectrum in high-density conditions is expected, physically interpretable, and consistent with prior analyses of congested-flow dynamics.

---

> > ### Author Response · Authors · 2025-11-28
> >
> > Dear Reviewer,
> >
> > We sincerely thank you for your constructive feedback. Following your suggestions, we have substantially improved the clarity and empirical rigor of the paper. New version of full manuscript is uploaded.
> >
> > In this version, we further conducted a systematic study of KDE-estimated density labels and find one unexpected observation which uncovers a in-depth mechanism. We use different KDE bandwidths of {10,20,25,30,40,50}. The trend aligns with our expectations that bandwidth of 25 meters performs best, an overly small or large bandwidths degrade accuracy but not exceed the advection-only ablation. One unexpected observation is that bandwidth of 10 meters performs even worse than the advection-only ablation. We carefully investigated this phenomenon and found that such a small bandwidth introduces high-frequency noise in KDE labels, which misguides the learning of $W^{\text{diff}}$, leading the commutator $L_{\text{JAD}}$ and spectral alignment of $L_{\text{spec}}$ disrupted. This result reveals an important insight: diffusion improves macro prediction only when the density supervision carries physically meaningful gradients; otherwise, diffusion becomes a harmful channel that injects noise into operator learning. The detailed analysis is included in the main paper.
> >
> > Following your recommendation, we described the intent label generation in Sec.5.2. The intent labels in our experiment are derived from the vehicle-level kinematic signals, following a deterministic procedure rooted in longitudinal acceleration patterns and lane-change dynamics. Notably, intent labels are only used while training for macro module supervision but not required when inference, therefore, the inference of our model is not suffered from it. We also test the runtime efficiency and provide the results in Appendix. Averaged over 10 runs with batch size 128, the inference time is 0.20 seconds, which supports real-time deployment.
> >
> > Our thoughts and answer for the other questions are provided in earlier comments. We sincerely appreciate your helpful suggestions, which have strengthened both the presentation and empirical analysis. If you have additional comments, we would be grateful for further discussion.

---

### Official Review · Reviewer_EX1y · 2025-10-31

**Soundness:** 3
**Presentation:** 3
**Contribution:** 3
**Rating:** 6
**Confidence:** 2

**Summary:**

The paper introduces Micro–Macro Coupled Koopman Modeling (MMCKM), a unified Koopman-based framework that jointly models microscopic vehicle interactions and macroscopic traffic flow evolution.

Unlike conventional microscopic or macroscopic approaches that treat these scales separately, MMCKM lifts the coupled dynamics into a shared high-dimensional linear observation space. On the macroscopic side, the method discretizes advection–diffusion traffic flow PDEs onto a vehicle-centric dynamic graph, preserving physical flow consistency without grid constraints. On the microscopic side, scenario-adaptive Koopman evolvers, guided by an Intent Discriminator, model diverse vehicle behaviors, while a Koopman control module captures the bidirectional influence between flow states and vehicle dynamics. MMCKM adopts a vehicle-centric dynamic graph that preserves microscopic perturbations while respecting macroscopic conservation laws by discretizing PDEs onto this graph. Evaluated on NGSIM and HighD datasets, MMCKM achieves comparable trajectory prediction accuracy to state-of-the-art history-dependent models despite using only real-time inputs. Ablation studies confirm the contributions of intent inference, macro-to-micro control, and operator interval design. The authors also outline future work on interpretable interaction analysis and extending the framework to urban traffic scenarios.

**Strengths:**

- The introduced method seems to be very original;
- The article is well-written; it has a good structure, and I like the fact that it includes an ablation study and a reproducibility statement, and the use of LLMs statement
- The results of the experiments are good
- The method has the potential to be significant, taking into account that it does not require historical trajectories

**Weaknesses:**

- The method still does not outperform the best methods utilizing historical data, but the fact that the results are comparable is noticeable
- There are some typos and minor writing issues, e.g., due to the lack of space,:
  - l. 49: "frameworksHuang" -> "frameworks Huang"
  - l. 50: "limitsCristiani" -> "limits Cristiani"
  - l. 128: "applicationsBrunton" -> "applications Brunton"
  - l. 133: "R^nLusch" -> "R^n Lusch"
  - l. 144: "spaceProctor" -> "space Proctor"
  - l. 370: "3 8 s" -> "3-8 seconds"

**Questions:**

-

---

> ### Author Response · Authors · 2025-11-20
>
> Thanks for reviewer providing such kindly suggestions.
> # Weakness 1
> We appreciate the reviewer’s observation, and we agree that methods using long historical sequences naturally have access to more information and may therefore achieve stronger short-term forecasting performance. In this sense, it is expected that our approach operating strictly from the current microscopic and macroscopic state may not surpass the best history-based models. The main contribution of our work is not to compete with sequence-dependent predictors, but to introduce a unified, theoretically grounded micro-macro dynamical framework that couples microscopic vehicle states with PDE-based macroscopic flow through Koopman operators. That our method achieves comparable performance despite using substantially less temporal information highlights the expressiveness of the underlying formulation.
>
> More importantly, the framework yields structured operators $C_{\text{adv}}$ and $L_{\text{diff}}$ that carry clear physical meaning and are directly connected to advection and diffusion mechanisms in traffic flow. These operators provide interpretable and potentially actionable signals for downstream tasks such as control, stability analysis, and simulation. We view the present work as a first step toward building such a principled multi-scale modeling tool, and we will continue to refine and extend this framework in future research.
>
> We have revised the manuscript to further clarify the contributions of our approach in light of the reviewer’s comment.
>
> # Weakness 2
> We also thank the reviewer for pointing out the typos and minor writing issues. We have carefully revised the manuscript and correct all formatting and spelling inconsistencies for the camera-ready version. Thanks again for these kind mention.

---

> > ### Comment · Reviewer_EX1y · 2025-11-26
> >
> > I appreciate the author's response to my review and to the comments of the other reviewers. For now, I have decided to keep my score.

---

> > > ### Author Response · Authors · 2025-11-28
> > >
> > > Dear Reviewer:
> > >
> > > Thanks for your kind response. We appreciate your positive evaluation and valuable feedback on our work.
> > >
> > > In line with your suggestions, we have uploaded a revised version of the full manuscript. In this revision, we have included a more in-depth investigation and discussion on the identified issues and the underlying mechanisms.
> > >
> > > Should you have any further questions or require additional clarification regarding this work, we welcome the opportunity for further discussion.

---

### Official Review · Reviewer_dPBn · 2025-11-01

**Soundness:** 3
**Presentation:** 3
**Contribution:** 4
**Rating:** 6
**Confidence:** 2

**Summary:**

The paper introduces Micro-Macro Coupled Koopman Modeling (MMCKM), a framework for unified traffic flow prediction that combines microscopic vehicle interactions and macroscopic flow dynamics by using Koopman operator theory. This combines microscopic dynamics, where every single vehicle is simulated, to macroscopic dynamics where the overall traffic flow is evolved. The method predicts the future state from the current time step, without relying on historical trajectories, by leveraging a high-dimensional linear observation space, and performs better than history-dependent baselines on experiments on NGSIM and HighD datasets.

**Strengths:**

The paper tackles an important problem, to bridge the gap between microscopic and macroscopic dynamics in traffic modeling, both of which have advantages and drawbacks. For instance the microscopic scale allows for simulating individual vehicle dynamics while the macroscopic scale enforces high-level flow dynamics such as conservation laws. The authors use an innovative Koopman-based unified model to jointly learn both scales, and leveraging a graph discretization of the PDE, as opposed to standard grid discretization.  Empirical validation on two real-world datasets (NGSIM & HighD) support the approach, achieving competitive or superior results without relying on historical trajectories. The ablation studies validate the impact of each module. The paper is well-written overall and well-motivated.

**Weaknesses:**

- The paper is quite technical and math-heavy, which can make it hard for readers not familiar with the area to follow the core concepts. Some sections such as 4.1/4.2 could benefit from more intuitive explanations or visual illustrations.
- Experiments are limited to highway datasets with estimated (not sensor) density labels, so it’s unclear how well the method would generalize.
- There are no direct comparisons with other hybrid or Koopman-based approaches, so it makes it harder to tell how much of the improvement comes from the coupling design itself.

**Questions:**

- Is the model applicable to urban or more complex scenarios than structured highways?
- Can the authors comment on the runtime efficiency of the method? Is it able to run at real-time?

---

> ### Author Response · Authors · 2025-11-20
>
> Thanks for reviewer providing such profound insights. We provide our thoughts in the following:
> # Weakness 1
> We thank the reviewer for this helpful suggestion.
> In the camera-ready version, we will include concise introductory figures and explanations in the main text that summarize the key ideas from Appendix A.1 and visually illustrate the concepts used in Section 4.1 and 4.2. These additions will make the intuition behind the derivations clearer.
> %We agree that, without reading Appendix A.1 in advance, the motivation and theoretical structure of Section 4.1 and 4.2 can indeed difficult to follow. Appendix A.1 contains a substantial amount of traffic-flow formulations and mathematical derivations, which makes the core ideas less accessible when only reading the main text.
>
> # Weakness 2
> We appreciate the reviewer's comment about this point. To best of our knowledge, there is no public dataset that simultaneously provides microscopic trajectories and sensor-measured density fields. This is why we construct macro density labels estimated by KDE through microscopic trajectories.
>
> More importantly, our proposed framework is not limited to highways or to any particular sensor setup: as long as microscopic trajectories are available, the same KDE-based aggregation can be used to obtain training labels, and after training the model, it can outputs local density from micro states, without requiring ground-truth density measurements. We are currently running sensitivity experiment with varying bandwidths. We aim to clearly illustrate the robustness of our framework under different bandwidth.
>
> The primary challenge in extending our approach from highway settings to more complex scenarios such as urban networks does not stem from the micro–macro formulation itself. Instead, it arises from the additional heterogeneity inherent to urban traffic such as interactions with pedestrians and cyclists, traffic signals and phase timing, intersection priority rules, multi-directional maneuvers and rule-level factors. These complex interactions significantly change the interaction graph. Extending our framework will therefore require a heterogeneous graph representation that incorporates vehicles, traffic signals and rule nodes. We are actively exploring this direction as part of our future work, and we have added the relevant discussion in the conclusion of the paper.
>
> In this submission, we intentionally focus on highways, so that the core contribution can be studied in a clean and controlled setting.
>
> # Weakness 3
> We appreciate the reviewer's concern. In fact, we find very few hybrid framework in the literature that jointly models microscopic vehicle trajectories and macroscopic traffic density evolution with a unified formulation. Most Koopman-based methods typically operate at a single scale thus do not provide directly comparable baselines for our multi-scale setting. On the other hand, existing macroscopic datasets and models focus on network-level densities. However, the factors that really influence vehicle's decisions are local, vehicle-centric for example the traffic density within perception range. Our goal is to investigate this density but there is lack of related sensor collected data or baselines.
>
> To assess how much improvement truly comes from the coupling mechanism itself, we included targeted ablations in which the macro influence is removed (MMCKM-C). The performance drops are substantial: short-horizon decreases by 40% and long-horizon by 27%. This indicates that the gains arise from the proposed cross-scale coupling.
>
> # Question 1
> We agree with the reviewer about the idea to extending the framework to urban or more complex scenarios. As we discussed in Weakness 2, we may need to utilize heterogenous graph to depict the scenario and we are currently investigating.
>
> In the present submission, we focus on structured highways because a purely vehicle-centric dynamic graph fully captures the relevant interactions, allowing us to clearly study and evaluate the proposed micro–macro Koopman framework. We will clarify this limitation and the implications for urban generalization in the final version.
>
> # Question 2
> We appreciate the reviewer's question regarding runtime efficiency and in fact, this is one of our advantages compared to existing spatiotempral model. After training, the inference pipeline is lightweight with only computing the following components once:
> * Macro-encoder GNN $O(E\cdot d)$ and micro-encoder MLP $O(d)$
> * Koopman evolver with control: $O(d^2)$
> * Decoder MLP $O(d)$
>
> All these operations are one-step and history-free, thus compared with RNN or Transformer-based predictors that process long historical trajectories and full self-attention, our model has substantially lower computational and memory cost thereby naturally suitable for real-time inference.
>
> We will include empirical runtime measurements in the camera-ready version to further illustrate this efficiency advantage.

---

> > ### Author Response · Authors · 2025-11-28
> >
> > Dear Reviewer:
> >
> > We sincerely thank you for the constructive feedback. Following your suggestions, we have substantially improved the clarity and empirical rigor of the paper.
> >
> > # Weakness 1
> > In response to your first concern, we add Fig.2 in the main text to explicitly illustrate how continuous traffic flow PDE formulation is converted onto a Vehicle-centric Lagrangian Graph. This figure visualizes the discretization process and highlights how advection and diffusion terms arise on the graph. We believe this addition makes the modeling assumptions more transparent and easier to follow.
> >
> > # Weakness 2
> > As described in earlier comment, there is no available sensor collected dataset for our simultaneous micro-macro architecture, we further conducted a systematic study of KDE-estimated density labels using bandwidth {10,20,25,30,40,50}. The results are collected in Table.5. The trend aligns with our expectations that bandwidth of 25 meters performs best, an overly small or large bandwidths degrade accuracy but not exceed the advection-only ablation. One unexpected observation is that bandwidth of 10 meters performs even worse than the advection-only ablation. We carefully investigated this phenomenon and found that such a small bandwidth introduces high-frequency noise in KDE labels, which misguides the learning of $W^{\text{diff}}$, leading the commutator $L_{\text{JAD}}$ and spectral alignment of $L_{\text{spec}}$ disrupted. The detailed analysis is included in the main paper.
> >
> > # Question 2
> > Following your recommendation, we now provide a runtime evaluation in the Appendix. Averaged over 10 runs with batch size 128, the inference time is 0.20 seconds, which supports real-time deployment.
> >
> > Our thoughts and answer for the other questions are provided in earlier comments. We sincerely appreciate your helpful suggestions, which have strengthened both the presentation and empirical analysis. If you have additional comments, we would be grateful for further discussion.

---

### Meta-Review · Area_Chair_E3hb · 2026-01-06

**Summary:**

This paper proposes Micro–Macro Coupled Koopman Modeling (MMCKM), a unified, physics-informed framework that couples microscopic vehicle dynamics with macroscopic traffic flow via Koopman operator theory.
The work is original and sound. The reviewers all seem to agree that this paper is above the bar.
The work offers a principled contribution to multi-scale traffic modeling, achieving comparable results.
The authors seeem to have addressed reviewers' concerns.

**Reviewer Concerns:**

The authors responded thoroughly to reviewer concerns by improving clarity with figures, adding KDE bandwidth sensitivity analysis, runtime evaluation, and clarifying intent labeling and learning procedures.

Weaknesses remain around limited experimental scope (primarily highway settings), reliance on KDE-estimated macro labels rather than sensor ground truth, and lack of direct comparison with other hybrid or Koopman-based multi-scale methods.
However I believe this paper is above bar the acceptance quality required by ICLR.

I strongly recommend the authors to revise the paper further by the camera ready deadline to further improve clarity and impact.

**Reviewer Scores:**

I believe 1-2 reviewers may have increased their scores. The original scores are all 6.

---

### Decision · Program_Chairs · 2026-01-26

Accept (Poster)